# How Does the Archaellum Work?

**DOI:** 10.3390/biom15040465

**Published:** 2025-03-21

**Authors:** Morgan Beeby, Bertram Daum

**Affiliations:** 1Department of Life Sciences, Imperial College London, London SW7 2AZ, UK; 2Living Systems Institute, University of Exeter, Exeter EX4 4SB, UK

**Keywords:** archaella, archaeal flagella, molecular machines, archaea, propulsive nanomachines

## Abstract

The archaellum is the simplest known molecular propeller. An analogue of bacterial flagella, archaella are long helical tails found in Archaea that are rotated by cell-envelope-embedded rotary motors to exert thrust for cell motility. Despite their simplicity, however, they are less well studied, and how they work remains only partially understood. Here we describe four key aspects of their function: assembly, the transition from assembly to rotation, the mechanics of rotation, and how rotation generates thrust. We outline future research directions that will enhance our understanding of archaellar function.

## 1. Introduction

Naturally occurring propellers have fascinated scientists for decades, evoking human-made machines, and representing elegant solutions for cellular propulsion [1]. While Eukaryotic cilia and Bacterial flagella have been intensely studied for decades; however, the propeller used by Archaea, the archaellum, remains relatively poorly understood.

Archaella feature a rotary motor embedded in the cell envelope that gyrates a multi-micron extracellular filament to generate thrust for cell propulsion. Although morphologically analogous to bacterial flagella, which are composed of 25 proteins and whose motors are over 50 nm wide, archaella are substantially simpler. Indeed, the simplest and best studied, the archaellum from *Sulfolobus acidocaldarius*, is composed of only seven proteins (Figure 1) [1,2,3].

Archaella are members of the type IV filament superfamily [4,5,6], which includes type II secretion systems and various type IV pili. Archaella and type IV filament superfamily members share a common assembly pathway. Initially, pilin subunits are expressed as membrane proteins via the SEC pathway. Cleavage of an N-terminal signal peptide by a pre-pilin peptidase then primes the pilins for their incorporation into a nascent pilus, which is produced by an assembly machinery [1].

For archaella, the core proteins of this machinery include a transmembrane assembly platform ArlJ (homologs include PilC or PilG in type IVa pili, and GspF in type II secretion systems), a cytoplasmic ATPase ArlI (homologs include PilB or PilF in type IVa pili and GspE in type II secretion systems; there are also paralogs associated with pilus retraction named PilT and PilU), and a structural component of the archaellar filament, one or more copies of proteins from the archaellin family (if there is a single copy, it is usually named ArlB). These three core protein families are universal to type IV filament superfamily members and together anchor and assemble an extracellular filament, which, in the case of the archaellum, subsequently acts as a propeller with the aid of additional proteins. There is no reason to believe that these core proteins function differently to assemble in different type IV filament superfamily members, and so what holds true for the assembly of type IV pili and type II secretion systems most likely holds true for archaella.

The similarities end with the above-mentioned, however. Archaellar filaments, unlike the extracellular filaments of other type IV filament superfamily members, rotate to form helical propellers for cell propulsion. Correspondingly, archaella feature proteins not found in other type IV filament superfamily members. In the cytoplasm, ArlH docks beneath ArlI, and putative stator complexes composed of ArlFG likely bridge from the plasma membrane to the S-layer, anchoring the stator part of the archaellar motor to the cell ultrastructure. Bridging between the cytoplasmic components and stator components are cytoplasmic ring proteins ArlX or ArlCDE, depending on species. Being unique to archaella, ArlH, ArlFG, and the cytoplasmic ring proteins have thus been implicated in archaellar rotation.

Here we discuss what is known about assembly, the subsequent switch to rotation, rotation itself, and how rotation generates propulsion.

## 2. The Archaellum Likely Assembles Using the Same Mechanism as Other Type IV Filament Superfamily Members

The archaellum is a member of the widespread type IV filament superfamily [4,6]. All type IV filament superfamily members self-assemble a membrane-associated structure that then actively assembles an extracellular filament from a pool of monomeric proteins in the membrane. Lengths of the filaments range from stub-like pseudopili in type II secretion systems—that may never extend beyond the periplasm—to filaments several microns long [7]. While there may be nuanced differences between superfamily members, their homology and conserved core proteins give no reason to believe that the fundamentals of filament assembly differ between family members.

The assembly of the motor complex begins with the transmembrane platform protein ArlJ [8]. ArlI localises to the membrane via an N-terminal helical bundle [8,9] and subsequently is recruited by a cytoplasmic region of ArlJ [9]. Next, ArlH docks to ArlI, followed by assembly of the cytoplasmic ring proteins [10,11]. Finally, the putative stator complexes formed of ArlFG assemble and bind the S-layer [10,12,13].

The assembled machinery subsequently harnesses ATP hydrolysis to assemble the archaellar filament from subunits of the archaellin family, which are diffusing in the plasma membrane [8,14,15]. Archaellin subunits consist of an N-terminal membrane-embedded ⍺-helix and a soluble β-strand-rich, globular C-terminus. Once an archaellin has been incorporated into the membrane, guided by its positively charged N-terminal class III signal peptide [16,17,18,19,20], a pre-pilin peptidase cleaves its signal peptide. This primes it for its imminent incorporation into the nascent archaellar filament [21]. Filament assembly is believed to be mediated by ArlJ powered from beneath by the AAA+-ATPase ArlI; in archaella, ArlH, in complex with ATP, is also required to stimulate ArlI [11], although other type IV filament superfamily members assemble despite lacking an ArlH homolog. It is curious that this differs from other type IV filament superfamily members, which assemble despite absence of ArlH.

Studies of the ATPases from archaella, type IVa pili, and type IVc pili (the latter previously referred to as Tad pili) have suggested a catalytic cycle in which the conformational states in monomers rotates with each cycle [22,23], powering filament assembly. Structures of ArlI and homologs reveal cyclic hexamers stretched to form twofold-symmetric ovals measuring 14 nm along their long axis, with nucleotide binding sites between subunits [14,24,25]. Monomers cycle from an open state to an ATP-bound pre-hydrolysis closed state to an ADP-bound post-hydrolysis closed state (and then release ADP and return to the open state) [22], consistent with evidence for ArlI hexamers binding two ATPs [11]. This cycle, viewed from outside the cell, features sequential 60° clockwise rotations of conformational states.

This rotation of conformations may impose upon ArlJ to drive archaellin monomers into the base of the filament. Consistent with a conserved mechanism across type IV filament superfamily members, conserved ArlJ regions interact with ArlI [26]. The different conformations of the nominally planar cyclic ATPase hexamer feature out-of-plane movements, which, in type IVa pili, have been suggested to push up against the ArlJ homolog, opening a space in the base of the filament into which an archaellin monomer can insert. That this space would open at 60° clockwise offsets for each cycle is entirely consistent with the ArlJ transmembrane platform reorienting by rotating 60° each round to add the next monomer of a right-handed filament. Although purified structures of the cytoplasmic domain of ArlJ homologs dimerise [27], however, contemporary modeling results of full-length proteins rather suggest trimerization [23,28]. It was previously suggested that a dimer of ArlJ family proteins might physically rotate, driven by the conformational rotation of the ArlI homolog [22]. Alternatively, if transmembrane platform proteins form trimers, it has been speculated that they act similarly to the chuck from a drill, clamping around the base of a pilin and lifting it out of the membrane upon ATPase action [28]. Regardless of the mechanism, ArlJ inserts new archaellin monomers into the base of the growing archaellar filament [26,29,30,31]. In situ structures will be critical to understand how ArlI and ArlJ family members interact to drive filament extension.

## 3. The Sole ATPase Must Switch from Assembly Function to Rotational Function Once the Archaellar Filament Is Fully Formed

ArlI is the only archaellar protein capable of ATP hydrolysis, and no archaellar proteins have been implicated as ion motive force transducers. In the absence of other candidates, therefore, it is generally believed that ArlI must power both assembly and rotation. Indeed, consistent with this, deletion of the extreme N-terminus of ArlI abolishes rotation but not assembly [8].

The archaellar filament grows to a length of several microns before the machinery switches from filament elongation to filament rotation. The chief candidate for triggering the switch is ArlH, in part due to homology to circadian clock protein KaiC implicating it as a timer [11,32]. This suggests that archaellar length is dictated by an ArlH timer mechanism instead of a molecular ruler mechanism. A hexamer of ArlH, which binds ArlI in the presence of nucleotides [33], has reduced ArlI affinity upon mutation of the nucleotide binding site [11], and features Walker-type ATP-binding motifs (albeit with a divergent Walker B motif). Indeed, ATP binds to a region of ArlH that, in homologs, has implications in oligomerisation [11].

Nevertheless, ArlH has never been shown to hydrolyse ATP [11,34], but rather, like KaiC, autophosphorylates [33]. Phosphorylated ArlH hexamers monomerise, and have reduced ArlI affinity. ArlH has also been shown to bind the cytoplasmic ring proteins [11,35], prompting two models for how it might regulate the switch from assembly to rotation [36]. In the first model, unphosphorylated ArlH facilitates ArlI–ArlJ interaction, driving archaellar filament assembly. Phosphorylation leads to dissociation of ArlH from ArlI, halting assembly, and permitting rotation, although ArlH would remain bound to the cytoplasmic ring proteins. This model opens the possibility that reversal of ArlH phosphorylation might revert it to act as a ‘brake’ again, switching back from rotation to assembly. In the second model, unphosphorylated ArlH directly links ArlI from the cytoplasmic ring proteins, thus preventing rotation but permitting assembly. Both models, however, appear incompatible with the fact that all other type IV filament superfamily members assemble just fine without ArlH or the cytoplasmic ring proteins. Structures of archaella in situ both pre- and post-switching will be required to differentiate these models.

## 4. Rotation Is Driven by the Same ATPase That Assembles the Archaellum

How archaella rotate still rests heavily upon speculation. Here we describe what clues currently exist, and what more is needed to understand the mechanism.

Clear clues come from the occurrences of different protein families. The correspondence of the presence of ArlH, the cytoplasmic ring proteins, ArlF, and ArlG with rotatory function strongly implicates that these proteins are directly involved in archaellum rotation. It is nevertheless unclear if ArlH is actively involved in rotation or just the switch from assembly. Although ArlH is required for archaellar filament assembly [37], all other type IV filament superfamily members assemble filaments without an ArlH homolog. Whether it is an active player in rotation, or required only for remodeling the apparatus for rotation, and subsequently uninvolved, remains to be understood. Together this hints that the default function of ArlI is rotation, and ArlH may initially impose upon ArlI to fulfil its ancestral assembly function before ArlH dissociates and ArlI asserts its now-default rotational functionality. This model does not, however, explain why ArlH would remain in contact with the cytoplasmic ring proteins [35]. As for understanding switching, what is needed is to understand the architectural role of ArlH in archaellar motors in situ both pre- and post-switch from assembly to rotation.

ArlF and ArlG are likely to form the “stator” component of the motor, against which the rotary components push. The two are divergent paralogous duplications of archaellins that retain the ability to form a helical filament, albeit less tightly packed than archaellar filaments: whereas archaellar filaments can be thought of as a triple helix of archaellin filaments, the ArlG filament resembles a single filament from a triple helix [38]. It is likely that ArlF caps an ArlG filament and binds to the S-layer, thus statically anchoring to the cell superstructure [13]. Indeed, mutation of ArlF’s S-layer binding ability, or deletion of ArlF or ArlG, abolishes archaellar motility [13]. Identifying the location of ArlF and ArlG in the archaellum in situ is critical to verifying this and understanding how they interact with other components.

Finally, the role of the cytoplasmic ring proteins remains relatively poorly defined. ArlX forms a ~30 nm diameter ring in the cytoplasm, tethered to the membrane by a single transmembrane domain [39]. ArlX interacts with ArlH and ArlI, and is stabilised by ArlJ, suggesting that it forms a ring component around the core machinery [10,39]. Curiously, ArlX is required for archaellation, suggesting either a direct role in archaellar filament or assembly, or newly evolved dependence of the core assembly machinery upon ArlX. ArlCDE, on the other hand, do not have a membrane tether. They interact with ArlH and have been implicated as involved in chemotaxis [35].

Other clues to understanding the mechanisms underlying how any machine works are characteristics of their mechanical output. Over the past decade, biophysical studies have revealed that the motor can spin both clockwise and counterclockwise (but does not extend or retract while rotating) [40,41]; steps of both 36° and 60° are evident in high frame-rate videos; and the torque output archaella (from of *Halobacterium salinarum*) is 160 pN⋅nm [42], which would require the free energy of hydrolysis of 12 ATPs (or more, if the motor is less than 100% efficient), which might be most simply explained as two of the six ArlI subunits hydrolyses ATP per 60° turn [43]. There is no clear explanation for the 36° steps, however, and these critical hallmarks of mechanism must be understood to understand rotation.

So much so good: ArlFG are static; ArlJ clamps the filament and thus most certainly rotates. Where is the interface between the rotor and the stator? In other words, what rotates and what is static? What pushes against what to generate rotation? The 60° steps in rotation were previously rationalised by evoking the happy coincidence of the twofold symmetry of the oval ATPase with the twofold symmetry of the ATPase’s oval shape across all type IV filament superfamily members [22]. The difference, it was argued, is that the additional proteins found in the archaellum somehow harnessed the inherent relative rotation of ArlJ and ArlI family members to directly rotate the archaellar filament. In this model, rotation of the twofold symmetric oval shape (not rotation of the physical hexamer—just the conformational shape), lead to physical rotation of the twofold symmetric ArlJ to retain its interactions with the ATPase. The inference that ArlJ rotates thus rested upon the potentially incorrect assumption that it is a dimer.

Recent findings are not inconsistent with this model, but they are not as satisfying. With an ArlJ trimer, an unrotated ArlJ versus a 60°-rotated ArlJ would have equivalent stabilities, raising a question as to what drives the rotation. Furthermore, this model evokes the need for an anchor structure that binds ArlI to the stator complexes to facilitate rotation. ArlH’s autophosphorylation, however, involves disassociation from ArlI upon switching to rotation—the exact opposite of what might be needed for this model, where ArlI might use ArlH as part of an anchor structure bridging to ArlFG.

Taken together, current evidence therefore suggests that ArlI is part of the rotor, along with ArlJ and the filament. ArlFG and the cytoplasmic ring proteins are the static components. And ArlH either locks the entire structure together to enable assembly before decoupling from ArlI to enable ArlI to rotate within the cytoplasmic ring. Indeed, ArlI may push against the cytoplasmic ring to generate rotation by co-opting conformational changes intrinsic to all type IV filament superfamily ATPases—the archaellum rotates while other type IV filament superfamily members do not because the others do not have a cytoplasmic ring to push against.

## 5. The Archaellum Is a Helical Propeller

The archaellar filament has been understudied relative to the other better-known examples of filament-powered propulsive nanomachines, the bacterial flagellum [44], and eukaryotic cilia [1,45,46] (here we use ‘cilia’ to conflate the eukaryotic superfamily of cilia and flagella organelles to avoid confusion with bacterial flagella). While not homologous, archaella are functionally analogous to the extracellular superhelical filaments of bacterial flagella, which generate thrust through rotary propulsion [1,47]. Eukaryotic cilia are complex membrane-lined macromolecular assemblies composed of a central pair of microtubules, surrounded by nine microtubular doublets [45,46]. Unlike archaella and flagella, which rotate their passive extracellular filament using envelope-embedded rotary motors, cilia ‘beat’ using molecular motors along their length [1,47]. Thousands of dynein motor proteins anchored to each doublet perform concerted power strokes on neighboring doublets, leading to a characteristic beating motion of the cilium. It is this beating motion of one or many cilia that generates thrust [45,46]. Although some cilia beat with a rotating waveform, the structure of the cilium itself does not rotate, and many beat with simpler beats such as propagating waves [48].

In bacteria, the torque generated by a membrane-spanning motor complex rotates the filament powered by an ion gradient across the cell membrane [49]. Counterclockwise rotation of the flagellum results in a forward motion of the bacterium, while clockwise rotation causes flagellar bundles to “fly apart” and a cellular tumbling motion, which reorientates the cell prior to another round of forward motion [49]. This pattern of swimming motion is characteristic of many motile bacteria, including species like *Escherichia coli* and *Salmonella*, allowing them to navigate their environment by alternating between straight swimming (run) and reorientation phases (tumble) in response to chemical gradients, a process known as chemotaxis [50].

Despite their independent origin and evolution, there are clear functional parallels between flagella and archaella. A key prerequisite for rotary propulsion by both archaella and flagella is their ability to form a stable corkscrew-like superhelix [51]. This becomes particularly apparent when one considers that at cellular-length scales, water acts like a highly viscous honey-like liquid [52]. The superhelical shape allows “boring” through the viscous medium to generate thrust efficiently, converting rotational motion into propulsion, much like a screw in a dense fluid. Like the flagellum, the archaellum is an extracellular filament that generates thrust through gyration, powered by a membrane-spanning motor complex [1].

The swimming behaviour of archaea was characterised before the Archaea domain was officially defined. These early observations demonstrated that archaella function as rotating structures that facilitate both forward and backward movement in *Halobacterium salinarum* [40,53,54]. Clockwise rotation allows the cell to advance forward, whereas counterclockwise rotation drives it in the opposite direction. Meanwhile, the filament retains its chiral superhelical form regardless of the spinning direction [43,54].

Recent studies by us and others have provided high-resolution structural insights into the archaellar filament, reinforcing the notion that archaella are a product of divergent evolution from a molecular ancestor common with type IV filament superfamily members [55,56,57,58,59].

In bacterial flagella, which can undergo polymorphic transitions between different helical forms (ranging from left-handed to straight to right-handed superhelices), the archaellar filaments investigated so far appear to maintain a constant (right-handed) helicity regardless of the rotation direction [43,51,60] (Figure 2). However, as in archaea, far fewer species have been investigated regarding archaellar supercoiling, and it remains to be established if similar polymorphisms exist.

Archaella share the same structural blueprint as all other type IV filament superfamily members. They are composed of a helical assembly of tadpole-shaped subunits, consisting of an N-terminal ⍺-helical tail domain and a β-strand-rich globular head domain. The tails, originally the transmembrane anchor of the monomeric archaellin in the membrane, pre-assembly, are hydrophobic and bundle up to form the core of the filament. The heads, meanwhile, are hydrophilic and line the periphery of the archaellar filament [55,56,57,58,59].

Consistent with this, the tail domains are largely conserved, while the head domains show greater variability [59]. This variability may enable archaeal species to adapt their archaella to their unique lifestyle, swimming behaviour, and habitat [59]; however, further experimentation will be required to test this hypothesis. For example, some archaellin head domains vary in size and therefore determine the diameter of archaella. It is conceivable that this tunes the stiffness of the filaments. The stiffness, in turn, may influence the superhelical waveform adopted and thus the thrust that they generate within their aqueous environment upon gyration [59]. For example, long superhelical pitches (i.e., relatively few turns of the superhelix along its length) would be able to exert thrust effectively in low-viscosity environments, while shorter helical pitches would be necessary to bore effectively through viscous environments. The hypothesised relationship between archaellum thickness, their superhelical waveform, thrust and swimming behaviour remains to be tested by future research.

Between the head and tail exists a hinge region, which enables the heads to adopt different conformations with respect to the tail. This allows the archaellin subunits to adopt specific conformations depending on their position in the filament [61] and leads to supercoiling as a key prerequisite for swimming motility [51]. Not all archaellar filaments are composed of a single archaellin [2,60], although it remains unclear what benefits come from using multiple types of archaellin building blocks. Archaellar operons can encode up to seven archaellins [2]. In some cases, only one is highly expressed and forms the bulk of the filament [55,56,57]. The lower-expressed archaellins, referred to as minor archaellins, are assumed to form filament-capping structures or motor components [2]. It has been hypothesised that these minor archaellins may constitute motor components or filament-capping structures [2]; however, higher-resolution structural investigations are required to shed light on this open question.

In some cases, such as the archaellum of the methanogen *Methanocaldococcus villosus*, two archaellins are expressed in equal amounts and assemble into a complex heteropolymeric filament. The benefit of such an assembly is unknown, but it is possible that an observed asymmetric distribution of the two archaellins in the archaellum aids supercoiling [61].

A final poorly understood aspect of archaellar propulsion is the role of glycosylation, as many archaellar filaments are heavily glycosylated [55,56,57,58,59]. Structures of archaellar filaments solved so far display two to eight glycosylation sites per archaellin [55,56,57,58,59], with great variety in the composition and structure of glycans. Indeed, some species have evolved specialised glycosylation loops or subdomains to incorporate additional glycans or prominently expose them on the filament’s surface [59,61].

Archaellar glycosylation is important for motility, as glycan ablation drastically reduces swimming motility [62]. However, glycan ablation does not appear to impact proper assembly or even the conformation of archaellins within archaella [59,63]. Furthermore, glycan ablation led to an increased bundling of archaellar filaments in *Halobacterium salinarum* [63], potentially impeding archaella-driven motility. It follows that glycosylation plays a functional role, perhaps in facilitating the workings of the archaellar motor or in modulating the dynamics of archaellar rotation [59]. Further mutational studies in combination with comparative high-resolution structural analysis of archaellar motor complexes would provide deeper insights into the role of glycans in archaeal motility.

## 6. What More Is Needed to Understand How the Archaellum Works?

We need insights from a range of techniques: structural, biophysical, and genetic, to resolve some of the unknowns described above.

Perhaps the biggest question toward understanding rotation is to understand which proteins rotate and which remain static. To this end we need to develop complete molecular models of the archaellum in situ, ideally including snapshots of states pre- and-post switching from assembly to rotation. This will be invaluable in visualising which proteins bind to which, and which feature interfaces that might be loose enough to facilitate sliding during rotation. Given the power of contemporary cryoEM it is conceivable that structures to a sufficient resolution to resolve secondary structural elements (<8 Å-resolution) be determined with the use of high-throughput electron cryo-tomography data acquisition schemes [64,65] and modelled using recent advances in structure prediction [66,67].

Interpreting these above-mentioned structures will require more insights from biophysical experiments. Increasing the temporal and spatial resolutions of measurements may provide insights into energetics beyond steps alone, including shapes of the energy landscape of rotation.

## 7. Conclusions

Despite substantial advances in recent years, archaellar have yet to yield many of the secrets behind how they work. In the coming years, the combination of in situ structures, high temporal resolution biophysics, targeted mutagenesis, and simulations are what is needed to gain access to these secrets.

## Figures and Tables

**Figure 1 biomolecules-15-00465-f001:**
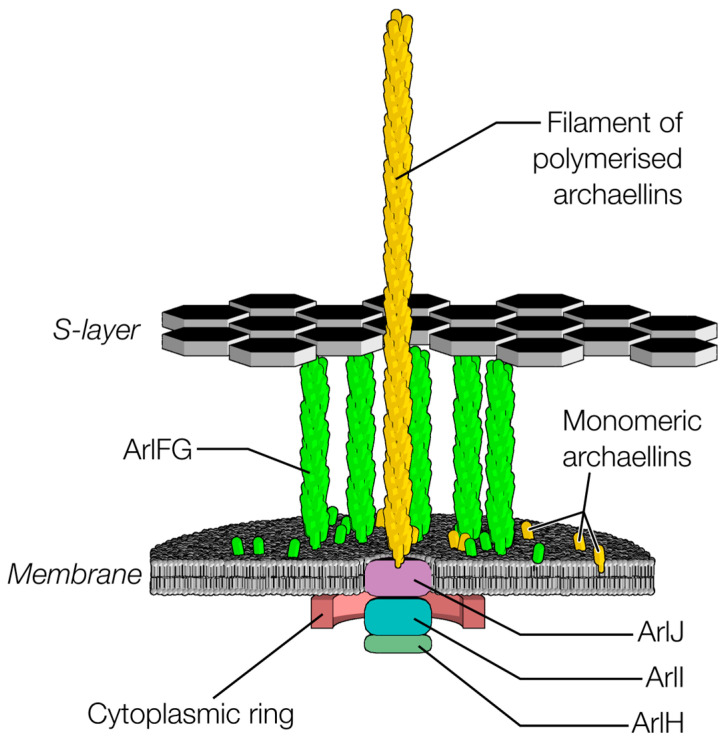
Schematic of the structure of the archaellum.

**Figure 2 biomolecules-15-00465-f002:**
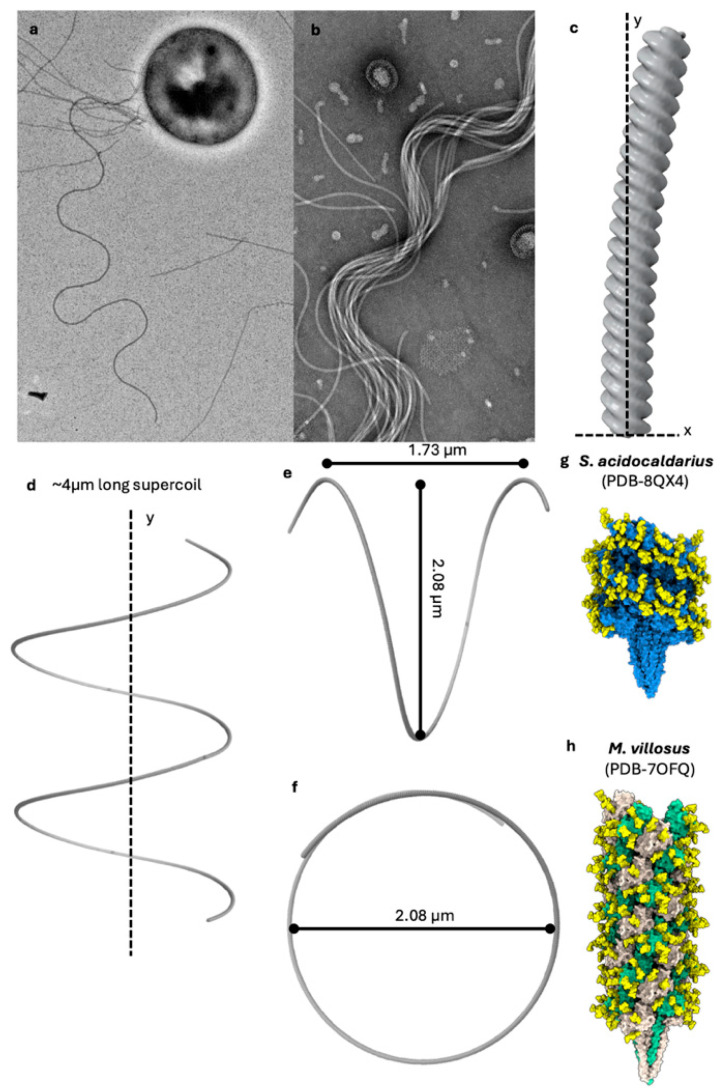
Archaellum filament structure and supercoiling. (**a**) Negative stain micrograph of a *Sulfolobus acidocaldarius* cell, with a supercoiled archaellum attached. (**b**) Bundle of archaella isolated from *S. acidocaldarius*. (**c**) Low-resolution single particle average of a curved section of an archaellum filament from *S. acidocaldarius*. (**d**) The curved filament extrapolated to a length of ~4 µm results in a superhelix with a pitch of 1.73 µm and a diameter of 2.08 µm (**e**,**f**). (**g**,**h**) Atomic models of the archaellum of *S. acidocaldarius* (**g**) and *Methanocaldococcus villosus* (**h**). The *S. acidocaldarius* archaellum consists of one repeating ArlB subunit (blue) (**g**). The *M. villosus* archaellum consists of two alternating component proteins, ArlB1 (grey) and ArlB2 (green) (**h**). Glycans are yellow.

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
