# Peer review of "How Does the Archaellum Work?"

_biomolecules, 2025, doi:10.3390/biom15040465_

Round 1

Reviewer 1 Report

Comments and Suggestions for Authors

The manuscript provides a brief overview of the current understanding of the structure and assembly of Archaeal flagella, as well as the structural basis of the functional mechanisms that underlie their rotation processes. The content is well-organized, featuring a clear, logical progression and concise language. It is an excellent review that is nearly ready for publication as it stands. I have some minor suggestions for the authors to consider:

Page 1, Line 22: “Although analogous to bacterial flagella, ” Suggestion: “Although morphologically analogous to bacterial flagella.”

Page 1, Line 25: …the archaellum from folobus acidocaldarius, is composed of only seven proteins (Fig. 1).” I suggest citing some literature or review papers here.

Page 3, Line 108.  “The sole ATPase must switch from assembly function to rotational function upon completion of the archaellar filament”. Suggestion:  “The sole ATPase must switch from assembly function to rotational function upon completion of the archaellar filament formation (or assembly)”

Page 4, Line 131: “This model opens the possibility reversal of ArlH phosphorylation might revert it to act as a ‘brake, switching back from rotation to assembly.’ Suggestion: “This model opens the possibility that, reversal of ArlH phosphorylation might revert it to act as a ‘brake’, switching back from rotation to assembly.’

Page 4, Line 134: “Both models, however, appear incompatible with all other type IV filament superfamily members assembling just fine without ArlH or the cytoplasmic ring proteins.” Suggestion: “Both models, however, appear incompatible with the fact that all other type IV filament superfamily members assemble just fine without ArlH or the cytoplasmic ring proteins.”

Page 4, Line 143: “.. the cytoplasmic ring proteins, and ArlFG strongly implicates them rotation” Suggestions: “the cytoplasmic ring proteins, and ArlFG strongly implicates that they rotate”

Author Response

Comment 1) Page 1, Line 22: “Although analogous to bacterial flagella, ” Suggestion: “Although morphologically analogous to bacterial flagella.”

Response: Fixed according to R1's suggestion.

Comment 2) Page 1, Line 25: …the archaellum from folobus acidocaldarius, is composed of only seven proteins (Fig. 1).” I suggest citing some literature or review papers here.

Response: We have added three reference as per R1's suggestion.

Comment 3) Page 3, Line 108.  “The sole ATPase must switch from assembly function to rotational function upon completion of the archaellar filament”. Suggestion:  “The sole ATPase must switch from assembly function to rotational function upon completion of the archaellar filament formation (or assembly)”

Response: Fixed according R1's suggestion.

Comment 4) Page 4, Line 131: “This model opens the possibility reversal of ArlH phosphorylation might revert it to act as a ‘brake, switching back from rotation to assembly.’ Suggestion: “This model opens the possibility that, reversal of ArlH phosphorylation might revert it to act as a ‘brake’, switching back from rotation to assembly.’

Response: Fixed according to R1's suggestion.

Comment 5) Page 4, Line 134: “Both models, however, appear incompatible with all other type IV filament superfamily members assembling just fine without ArlH or the cytoplasmic ring proteins.” Suggestion: “Both models, however, appear incompatible with the fact that all other type IV filament superfamily members assemble just fine without ArlH or the cytoplasmic ring proteins.”

Response: Fixed according to R1's suggestion.

Comment 6) Page 4, Line 143: “.. the cytoplasmic ring proteins, and ArlFG strongly implicates them rotation” Suggestions: “the cytoplasmic ring proteins, and ArlFG strongly implicates that they rotate”

Response: We fixed this as follows: “The correspondence of the presence of the cytoplasmic ring proteins, ArlH, ArlF, ArlG with rotatory function strongly implicates that these proteins are directly involved in archaellum rotation."

Reviewer 2 Report

Comments and Suggestions for Authors

The review “How does the archaellum work?” by Beeby and Daum gives an overview of the current knowledge of the archaellum, a phylogenetically unrelated analogue of the bacterial flagellum, belonging to the T4F superfamily. The manuscript is well written and relatively easy to follow, but could be improved. I have the following comments.

1) lines 9-10: although they have clearly been studied much less than bacterial flagella, I do not think archaella have been “relatively poorly-studied”. There is now an ample literature on these filaments.

2) line 17: “Naturally-occurring propellers fascinate us” is a bit of an overstatement. I am not sure everyone shares the authors (and mine) interest on this topic.

3) line 27: it is not clear to me what the authors mean by “adhesin variants”.

4) lines 27-33: an important universally conserved core protein is overlooked, the prepilin peptidase. Why? This needs to be corrected.

5) lines 30-31: perhaps delete the mention about “paralogs associated with pilus retraction named PilT and PilU”, which might confuse some readers and is not really crucial for this review.

6) An important general comment is that it would immensely help the readers follow the text if the authors referred to the proteins more often by their generic names rather than exclusively by their Arl nomenclature, e.g. state platform protein instead of ArlJ, ATPase motor instead of ArlI etc.

7) line 57: are they really “filaments longer than 20 μm”? If so, this is really an extreme case because most T4F are a few µm long.

8) lines 74-77: these two sentences are repeated, which is unnecessary and needs to be corrected

9) Another important general comment is that the figures could have been much better, and used to convey essential information. This is especially true for Figure 1, which is extremely basic. For example, the generic names of the different components should have been stated on Figure 1, which should have displayed the prepilin peptidase as well. Additionally, it would have been nice to finish on a model figure about how archaella might work.

10) lines 141-143: this sentence does not make sense.

11) line 230 and elsewhere: bacterial and archaeal names should be in italics.

12) legend to Fig. 2: please correct the typo “Methancaldococcus villosus”.

13) line 288: replace “for” by “form”, or better “constitute”, otherwise this sentence does not make sense.

14) line 329: what is meant by “synthesis of in situ structures”?

Author Response

Comment 1) lines 9-10: although they have clearly been studied much less than bacterial flagella, I do not think archaella have been “relatively poorly-studied”. There is now an ample literature on these filaments.

Response 1) We have replaced "relatively poorly understood" with "less well understood". This is clearly the case, as less structural information is available on the archaellum motor compared to that of the bacterial flagellum.

Comment 2) line 17: “Naturally-occurring propellers fascinate us” is a bit of an overstatement. I am not sure everyone shares the authors (and mine) interest on this topic.

Response: We toned down this sentence as follows: "Naturally-occurring have fascinated scientists for decades, evoking human-made machines, and representing elegant solutions for cellular propulsion"

Comment 3) line 27: it is not clear to me what the authors mean by “adhesin variants”.

Response: Fair enough - we deleted "adhesion variants" and improved the sentence for clarity. 

Comment 4) lines 27-33: an important universally conserved core protein is overlooked, the prepilin peptidase. Why? This needs to be corrected.

Response: The prepilin peptidase and its action is now explained. 

Comment 7) line 57: are they really “filaments longer than 20 μm”? If so, this is really an extreme case because most T4F are a few µm long.

Response: We toned down the sentence by saying "several μm".

Comment 10) lines 141-143: this sentence does not make sense.

Response. We fixed the sentence according to R1's suggestion. 

Comment 11) line 230 and elsewhere: bacterial and archaeal names should be in italics.

Response: Fixed

Comment 12) legend to Fig. 2: please correct the typo “Methancaldococcus villosus”.

Response: Fixed

Comment 13) line 288: replace “for” by “form”, or better “constitute”, otherwise this sentence does not make sense.

Response: Fixed

Comment 14) line 329: what is meant by “synthesis of in situ structures”?

Response: we have replaced "synthesis" with "a combination", which reads more clearly.